# PRE-TRAINED CONTEXTUAL EMBEDDING OF SOURCE CODE

## ABSTRACT

The source code of a program not only serves as a formal description of an executable task, but it also serves to communicate developer intent in a human-readable form. To facilitate this, developers use meaningful identifier names and natural-language documentation. This makes it possible to successfully apply sequence-modeling approaches, shown to be effective in natural-language processing, to source code. A major advancement in natural-language understanding has been the use of pre-trained token embeddings; BERT and other works have further shown that pre-trained contextual embeddings can be extremely powerful and can be finetuned effectively for a variety of downstream supervised tasks. Inspired by these developments, we present the first attempt to replicate this success on source code. We curate a massive corpus of Python programs from GitHub to pre-train a BERT model, which we call *Code Understanding BERT* (CuBERT). We also pre-train Word2Vec embeddings on the same dataset. We create a benchmark of five classification tasks and compare finetuned CuBERT against sequence models trained with and without the Word2Vec embeddings. Our results show that CuBERT outperforms the baseline methods by a margin of 2.9–22%. We also show its superiority when finetuned with smaller datasets, and over fewer epochs. We further evaluate CuBERT's effectiveness on a joint classification, localization and repair task involving prediction of two pointers.

## 1 INTRODUCTION

Modern software development places a high value on writing clean and readable code. This helps other developers understand the author's intent so that they can maintain and extend the code. Developers use meaningful identifier names and natural-language documentation to make this happen (Martin, 2008). As a result, source code contains substantial information that can be exploited by machine-learning algorithms. Sequence modeling on source code has been shown to be successful in a variety of software-engineering tasks, such as code completion (Hindle et al., 2012; Raychev et al., 2014), source code to pseudocode mapping (Oda et al., 2015), API-sequence prediction (Gu et al., 2016), program repair (Pu et al., 2016; Gupta et al., 2017), and natural language to code mapping (Iyer et al., 2018), among others.

The distributed vector representations of tokens, called token (or word) embeddings, are a crucial component of neural methods for sequence modeling. Learning useful embeddings in a supervised setting with limited data is often difficult. Therefore, many unsupervised learning approaches have been proposed to take advantage of large amounts of unlabeled data that are more readily available. This has resulted in ever more useful pre-trained token embeddings (Mikolov et al., 2013a; Pennington et al., 2014). However, the subtle differences in the meaning of a token in varying contexts are lost when each word is associated with a single representation. Recent techniques for learning contextual embeddings (McCann et al., 2017; Peters et al., 2018; Radford et al., 2018; 2019; Devlin et al., 2019; Yang et al., 2019) provide ways to compute representations of tokens based on their surrounding context, and have shown significant accuracy improvements in downstream tasks, even with only a small number of task-specific parameters.

Inspired by the success of pre-trained contextual embeddings for natural languages, we present the first attempt to apply the underlying techniques to source code. In particular, BERT (Devlin et al., 2019) produces a bidirectional Transformer encoder (Vaswani et al., 2017) by training it to

predict values of masked tokens and whether two sentences follow each other in a natural discourse. The pre-trained model can be finetuned for downstream supervised tasks and has been shown to produce state-of-the-art results on a number of NLP benchmarks. In this work, we derive contextual embedding of source code by training a BERT model on source code. We call our model CuBERT, short for *Code Understanding BERT*.

In order to achieve this, we curate a massive corpus of Python programs collected from GitHub. GitHub projects are known to contain a large amount of duplicate code. To avoid biasing the model to such duplicated code, we perform deduplication using the method of Allamanis (2018). The resulting corpus has 6.6M unique files with a total of 2 billion words. We also train Word2Vec embeddings (Mikolov et al., 2013a;b), namely, continuous bag-of-words (CBOW) and Skipgram embeddings, on the same corpus. For evaluating CuBERT, we create a benchmark of five classification tasks, ranging from classification of source code according to presence or absense of certain classes of bugs, to mismatch between a function's natural language description and its body, to predicting the right kind of exception to catch for a given code fragment. These tasks are motivated by prior work in this space, but unfortunately, the associated datasets come from different languages and varied sources. We want to ensure that there is no overlap between pre-training and finetuning datasets, and that all of the tasks are defined on Python code. We therefore create new datasets for the five tasks after carefully separating the pre-training and finetuning corpora. To evaluate CuBERT's effectiveness on a more complex task, we create a task for joint classification, localization and repair of variable misuse bugs (Vasic et al., 2019), which involves predicting two pointers.

We finetune CuBERT on each of the classification tasks and compare the results with multi-layered bidirectional LSTM (Hochreiter & Schmidhuber, 1997) models. We train the LSTM models from scratch and also using pre-trainned Word2Vec embeddings. Our results show that CuBERT consistently outperforms these baseline models by 2.9–22% across the tasks. We perform a number of additional studies by varying the sampling strategies used for training Word2Vec models, by varying program lengths, and by comparing against Transformer models trained from scratch. In addition, we also show that CuBERT can be finetuned effectively using only 33% of the task-specific labeled data and with only 2 epochs, and that it attains results competitive to the baseline models trained with the full datasets and much larger number of epochs. CuBERT when finetuned on the variable misuse localization and repair task, produces high classification, localization and localization+repair accuracies. The contributions of this paper are as follows:

- We present the first attempt at pre-training a BERT contextual embedding of source code.

- We show the efficacy of the pre-trained contextual embedding on five classification tasks. Our results show that the finetuned models outperform the baseline LSTM models supported by Word2Vec embeddings, and Transformers trained from scratch. Further, the finetuning works well even for smaller datasets and fewer training epochs. We also evaluate CuBERT on a multi-headed pointer prediction task.

- We plan to make the models and datasets publicly available for use by others.

## 2    RELATED WORK

Given the abundance of natural-language text, and the relative difficulty of obtaining labeled data, much effort has been devoted to using large corpora to learn about language in an unsupervised fashion, before trying to focus on tasks with small labeled training datasets. Word2Vec (Mikolov et al., 2013a;b) computed word embeddings based on word co-occurrence and proximity, but the same embedding is used regardless of the context. The continued advances in word embeddings (Pennington et al., 2014) led to publicly released pre-trained embeddings, used in a variety of tasks.

To deal with varying word context, contextual word embeddings were developed (McCann et al., 2017; Peters et al., 2018; Radford et al., 2018; 2019), in which an embedding is learned for the *context* of a word in a particular sentence, namely the sequence of words preceding it and possibly following it. BERT (Devlin et al., 2019) improved natural-language pre-training by using a denoising autoencoder. Instead of learning a language model, which is inherently sequential, BERT optimizes for predicting a noised word within a sentence. Such prediction instances are generated by choosing a word position and either keeping it unchanged, removing the word, or replacing the word with a random wrong word. It also pre-trains with the objective of predicting whether two sentences

can be next to each other. These pre-training objectives, along with the use of a Transformer-based architecture, gave BERT an accuracy boost in a number of NLP tasks over the state-of-the-art. BERT has been improved upon in various ways, including modifying training objectives, utilizing ensembles, combining attention with autoregression (Yang et al., 2019), and expanding pre-training corpora and time (Liu et al., 2019). However, the main architecture of BERT seems to hold up as the state-of-the-art, as of this writing.

In the space of programming languages, attempts have been made to learn embeddings in the context of specific software-engineering tasks. These include embeddings of variable and method identifiers using local and global context (Allamanis et al., 2015), abstract syntax trees or ASTs (Mou et al., 2016), paths in ASTs (Alon et al., 2019), memory heap graphs (Li et al., 2016), and ASTs enriched with data flow information (Allamanis et al., 2018). These approaches require analyzing source code beyond simple tokenization. In this work, we derive a pre-trained contextual embedding of tokenized source code without explicitly modeling source-code-specific information, and show that the resulting embedding can be effectively finetuned for downstream tasks.

## 3 EXPERIMENTAL SETUP

### 3.1 CODE CORPUS FOR FINETUNING TASKS

We use the ETH Py150 corpus (Raychev et al., 2016) to generate datasets for the finetuning tasks. The ETH Py150 corpus consists of 150K Python files from GitHub, and is partitioned into a training split (100K files) and a test split (50K files). We held out 10K files from the training split as a validation split. We deduplicated the dataset in the fashion of Allamanis (2018), resulting in a slightly smaller dataset of 85K, 9.5K, and 47K files in train, validation, and test, respectively.

### 3.2 THE GITHUB PYTHON PRE-TRAINING CODE CORPUS

We used the public GitHub repository hosted on Google's BigQuery platform (the `github_repos` dataset under BigQuery's public-data project, `bigquery-public-data`). We extracted all files ending in `.py`, under open-source, redistributable licenses, removed symbolic links, and retained only files reported to be in the `refs/heads/master` branch. This resulted in about 16.1M files.

To avoid duplication between pre-training and finetuning data, we removed files that had high similarity to the files in the ETH Py150 dataset, using the method of Allamanis (2018). In particular, two files are considered similar to each other if the Jaccard similarity between the sets of tokens (identifiers and string literals) is above 0.8 and in addition, it is above 0.7 for multi-sets of tokens. This brought the dataset to 13.5M files. We then further deduplicated the remaining files, by clustering them into equivalence classes holding similar files according to the same similarity metric, and keeping only one exemplar per equivalence class. This helps avoid biasing the pre-trained embedding. Finally, we removed files that could not be tokenized. In the end, we were left with 6.6M Python files containing over 2 billion words. This is our Python pre-training code corpus.

### 3.3 SOURCE CODE MODELING

We first tokenize a Python program using the standard Python tokenizer (the `tokenize` package). We leave language keywords intact and produce special tokens for syntactic elements that have either no string representation (e.g., DEDENT tokens, which occur when a nested program scope concludes), or ambiguous interpretation (e.g., new line characters inside string literals, at the logical end of a Python statement, or in the middle of a Python statement result in distinct special tokens). We split identifiers according to common heuristic rules (e.g., snake or Camel case). Finally, we split string literals using heuristic rules, on whitespace characters, and on special characters. We limit all thus produced tokens to a maximum length of 15 characters. We call this the *program vocabulary*. Our Python pre-training code corpus contained 10.2M unique tokens, including 12 reserved tokens.

We greedily compress the program vocabulary into a *subword vocabulary* (Schuster & Nakajima, 2012) using the `SubwordTextEncoder` from the Tensor2Tensor project (Vaswani et al., 2018), resulting in slightly over 50K tokens. All words in the program vocabulary can be losslessly encoded using one or more of the subword tokens.

We encode programs first into program tokens, as described above, and then encode those tokens one by one in the subword vocabulary. The objective of this encoding scheme is to preserve syntactically meaningful boundaries of tokens. For example, the identifier "`snake_case`" could be encoded as "`sna ke _ ca se`", preserving the snake case split of its characters, even if the subtoken "`e_c`" were very popular in the corpus; the latter encoding might result in a smaller representation but would lose the intent of the programmer in using a snake-case identifier. Similarly, "`i=0`" may be very frequent in the corpus, but we still force it to be encoded as separate tokens `i`, `=`, and `0`, ensuring that we preserve the distinction between operators and operands. Both the BERT model and the Word2Vec embeddings are built on the subword vocabulary.

## 3.4 FINETUNING TASKS

To evaluate CuBERT, we design five classification tasks and a multi-headed pointer task. These are motivated by prior work, but unfortunately, the associated datasets come from different languages and varied sources. We want the tasks to be on Python code, and for accurate results, we ensure that there is no overlap between pre-training and finetuning datasets. We therefore create all the tasks on the ETH Py150 corpus (see Section 3.1). As discussed in Section 3.2, we ensure that there is no duplication between this and the pre-training corpus. We hope that our datasets for these tasks will be useful to others as well. The finetuning tasks are described below. A more detailed discussion is presented in Appendix A.

**Variable Misuse Classification**   Allamanis et al. (2018) observed that developers may mistakenly use an incorrect variable in the place of a correct one. These mistakes may occur when developers copy-paste similar code but forget to rename all occurrences of variables from the original fragment, or when there are similar variable names in contexts that can be confused with each other. These can be subtle errors that remain undetected during compilation. The task by Allamanis et al. (2018) is to predict a correct variable name at a location within a function and was devised on C# programs. We take the classification version restated by Vasic et al. (2019), wherein, given a function, the task is to predict whether there is a variable misuse at some location in the function, without specifying a particular location to consider. In this setting, the classifier has to consider all variables and their usages to make the decision. In order to create negative (buggy) examples, we replace a variable use at some location with another variable that is defined within the function.

**Wrong Binary Operator**   Pradel & Sen (2018) proposed the task of detecting whether a binary operator in a given expression is correct. They use features extracted from limited surrounding context. We use the entire function with the goal of detecting whether any binary operator in the function is incorrect. The negative examples are created by randomly replacing some binary operator with another type-compatible operator.

**Swapped Operand**   Pradel & Sen (2018) propose the wrong binary operand task where a variable or constant is used incorrectly in an expression, but that task is quite similar to the variable misuse task we already use. We therefore define another class of operand errors where the operands of non-commutative binary operators are swapped. The operands can be arbitrary subexpressions, and are not restricted to be just variables or constants. To simplify example generation, we restrict examples for this task to those in which the binary operator and its operands all fit within a single line.

**Function-Docstring Mismatch**   Developers are encouraged to write descriptive docstrings to explain the functionality and usage of functions. This provides parallel corpora between code and natural language sentences that have been used for machine translation between the two (Barone & Sennrich, 2017), detecting uninformative docstrings (Louis et al., 2018) and to evaluate their utility to provide supervision in neural code search (Cambronero et al., 2019). We prepare a sentence-pair classification problem where the function and its docstring form two distinct sentences. Similar to the other finetuning tasks, we use the ETH Py150 corpus to create this dataset. The positive examples come from the correct function-docstring pairs. We create negative examples by replacing correct docstrings with docstrings of other functions, randomly chosen from the dataset. For this task, the existing docstring is removed from the function body.

|                                          | Train   | Validation     | Test   |
|------------------------------------------|---------|----------------|--------|
| Variable Misuse Classification           | 796020  | 8192  (86810)  | 429854 |
| Wrong Binary Operator                    | 537244  | 8192  (59112)  | 293872 |
| Swapped Operand                          | 276116  | 8192  (30818)  | 152248 |
| Function-Docstring                       | 391049  | 8192  (44029)  | 213269 |
| Exception Type                           | 21694   | 2459    (2459) | 12036  |
| Variable Misuse Localization and Repair  | 796020  | 8192  (86810)  | 429854 |

Table 1: Benchmark finetuning datasets. Note that for validation, we have subsampled the original datasets (in parentheses) down to 8192 examples, except for exception classification, which only had 2459 validation examples, all of which are included.

**Exception Type**  While it is possible to write generic exception handlers (e.g., "`except Exception`" in Python), it is considered a good coding practice to catch and handle the precise exceptions that can be raised by a code fragment. We identified the 20 most common exception types from the GitHub dataset, excluding the catch-all `Exception` (full list in Table 6). Given a function with an `except` clause for one of these exception types, we replace the exception with a special "hole" token. The task is the multi-class classification problem of predicting the original exception type.

**Variable Misuse Localization and Repair**  As an instance of a non-classification task, we consider the joint classification, localization and repair version of the variable misuse task from Vasic et al. (2019). Given a function, the task is to predict one pointer (called the localization pointer) to identify a variable misuse location and another pointer (called the repair pointer) to identify a variable from the same function that is the right one to use at the faulty location. The model is also trained to classify functions that do not contain any variable misuse as bug-free by making the localization pointer point to a special location in the function. We create negative examples using the same method as used in the Variable Misuse Classification task.

Table 1 lists the sizes of the resulting benchmark datasets extracted from the (deduplicated) ETH Py150 corpus. The Exception Type task contains fewer examples than the other tasks, since examples for this task only come from functions that catch one of the chosen 20 exception types.

## 3.5  BERT FOR SOURCE CODE

The BERT model (Devlin et al., 2019) consists of a multi-layered Transformer encoder. It is trained with two tasks: (1) to predict the correct tokens in a fraction of all positions, some of which have been replaced with incorrect tokens or the special `[MASK]` token (the Masked Language Model task) and (2) to predict whether the two sentences separated by the special `[SEP]` token follow each other in some natural discourse (the Next Sentence Prediction task). Thus, each example consists of one (for MLM) or two (for NSP) *sentences*, where a sentence is the concatenation of contiguous lines from the source corpus, sized to fit the target example length. To ensure that every sentence is treated in multiple instances of both MLM and NSP, BERT by default duplicates the corpus 10 times, and generates independently derived examples from each duplicate. With 50% probability, the second example sentence comes from a random document (for NSP). With 15% probability, a token is chosen for an MLM prediction (up to 20 per example), and from those chosen, 80% are masked, 10% are left undisturbed, and 10% are replaced with a random token.

CuBERT is similarly formulated, but a CuBERT sentence is a logical code line, as defined by the Python standard. Intuitively, a logical code line is the shortest sequence of consecutive lines that may constitute a legal statement, e.g., it has correctly matching parentheses. We count example lengths by counting the subword tokens of both sentences (see Section 3.3).

We train the BERT Large model, consisting of 24 layers with 16 attention heads and hidden size of 1024 units. Sentences are created by parsing our pre-training dataset. Task-specific classifiers pass the embedding of a special start-of-example `[CLS]` token through feedforward and softmax layers. For the pointer prediction task, the pointer is computed over the sequence of outputs generated by the last layer of the BERT model.

### 3.6 BASELINES

#### 3.6.1 WORD2VEC

We train Word2Vec models using the same pre-training corpus as the BERT model. To maintain parity, we generate the dataset for Word2Vec using the same pipeline as BERT but by disabling masking and generation of negative examples for NSP. The dataset is generated without any duplication. We train both CBOW and Skipgram models using GenSim (Řehůřek & Sojka, 2010). To deal with the large vocabulary, we use negative sampling and hierarchical softmax (Mikolov et al., 2013a;b) to train the two versions. In all, we obtain four Word2Vec embeddings.

#### 3.6.2 BIDIRECTIONAL LSTM AND TRANSFORMER

In order to obtain context-sensitive encodings of input sequences for the finetuning tasks, we use multi-layered bidirectional LSTMs (Hochreiter & Schmidhuber, 1997) (BiLSTMs). These are initialized with the pre-trained Word2Vec embeddings. Additionally, to further evaluate whether LSTMs alone are sufficient without pre-training, we try initializing the BiLSTM with an embedding matrix that is trained from scratch. We also trained Transformer models (Vaswani et al., 2017) for our finetuning tasks. We used BERT's own Transformer implementation, to ensure comparability of results. For comparison with prior work, we use the unidirectional LSTM and pointer model from (Vasic et al., 2019) for the Variable Misuse Localization and Repair task.

## 4 EXPERIMENTAL RESULTS

### 4.1 TRAINING DETAILS

As stated above, CuBERT's dataset generation duplicates the corpus 10 times, whereas Word2Vec is trained without duplication. To compensate for this difference, we trained Word2Vec for 10 epochs and CuBERT for 1 epoch. We pre-train CuBERT with the default configuration of the BERT Large model. For sequences of length 128, 256 and 512, we use batch sizes of 8192, 4096 and 2048 respectively. For Word2Vec, when training with negative samples, we choose 10 negative samples. The embedding sizes for all the pre-trained models are set at 1024.

For the baseline BiLSTM models, we did extensive experimentation on the Variable Misuse task by varying the number of layers (1–3) and the number of hidden units (128, 256, 512). We also tried LSTM output dropout probability (0.1, 0.5), optimizers (Adam (Kingma & Ba, 2014) and Ada-Grad (Duchi et al., 2011)), and learning rates (1e-3, 1e-4, 1e-5). The most promising combination was a 3-layered BiLSTM with 512 hidden units per layer, LSTM output dropout probability of 0.1 and Adam optimizer with learning rate of 1e-3. We use this set of parameters for all the tasks except the Exception Type task. Due to the much smaller dataset size of the latter (Table 1), we did a separate search and chose a single-layer BiLSTM with 256 hidden units. We used the batch size of 8192 for the larger tasks and 64 for the Exception Type task. For the baseline Transformer models, we originally attempted to train a Transformer model of the same configuration as CuBERT. However, the size of our training dataset seemed too small to train that large a Transformer. Instead, we performed a hyperparameter search over transformer layers (1–6), hidden units (128, 256, 512), learning rates (5e-5, 1e-4, 5e-4, 1e-3) and batch sizes (64, 256, 1024, 2048, 4096, 8192) on the Variable Misuse task. The best architecture (4 layers, 512 hidden units, 16 attention heads, learning rate of 5e-4, batch size of 4096) is used for all the tasks except the Exception Type task. A separate experimentation for the smaller Exception Type dataset resulted in the best configuration of 3 layers, 512 hidden units, 16 attention heads, learning rate of 5e-5, and batch size of 2048.

Finally, for our baseline pointer model (referred to as LSTM+pointer below) we searched over the following hyperparameter choices: hidden sizes of 512 and 1024, token embedding sizes of 512 and 1024, learning rates of 0.1, 0.01, and 0.001, and the AdaGrad and Gradient Descent optimizers. In contrast to the original work, we generated one pair of buggy/bug-free examples per function (rather than one per variable use, per function, which would bias towards longer functions), use CuBERT's subword-tokenized vocabulary of 50K subtokens (rather than a limited full-token vocabulary, which leaves many tokens out of vocabulary).

| | Setting | | Misuse | Operator | Operand | Docstring | Exception |
|---|---|---|---|---|---|---|---|
| **BiLSTM** (100 epochs) | From scratch | | 76.05% | 82.00% | 87.77% | 78.43% | 40.37% |
| | CBOW | ns | **77.66**% | **84.42**% | 88.66% | **89.13**% | 48.85% |
| | | hs | 77.01% | 84.11% | **89.69**% | 86.74% | 46.73% |
| | Skipgram | ns | 71.58% | 83.06% | 87.67% | 84.69% | 48.54% |
| | | hs | 77.21% | 83.06% | 89.01% | 82.56% | **49.68**% |
| **CuBERT** | 2 epochs | | 90.09% | 85.15% | 88.67% | 95.81% | 52.38% |
| | 10 epochs | | 92.73% | 88.43% | 88.67% | 95.81% | 62.55% |
| | 20 epochs | | **94.61**% | **90.24**% | **92.56**% | **96.85**% | **71.74**% |
| **Transformer** | (100 epochs) | | 79.37% | 78.66% | 86.21% | 91.10% | 48.60% |

Table 2: Test accuracies of finetuned CuBERT against BiLSTM (with and without Word2Vec embeddings) and Transformer trained from scratch on the classification tasks. "ns" and "hs" respectively refer to negative sampling and hierarchical softmax settings used for training CBOW and Skipgram models. "From scratch" refers to training with freshly initialized token embeddings, that is, without pre-trained Word2Vec embeddings.

## 4.2 RESEARCH QUESTIONS

We set out to answer the following research questions. We will address each with our results.

1. Do contextual embeddings help with source-code analysis tasks, when pre-trained on an unlabeled code corpus? We compare CuBERT to BiLSTM models with and without pre-trained Word2Vec embeddings on the classification tasks (Section 4.3).

2. Does finetuning actually help, or is the Transformer model behind CuBERT the main power behind the approach? We compare finetuned CuBERT models to Transformer-based models trained from scratch on the classification tasks (Section 4.4).

3. How does the performance of CuBERT on the classification tasks scale with the amount of labeled training data? We compare the performance of finetuned CuBERT models when finetuning with one third, two thirds, or the full training dataset for each task (Section 4.5).

4. How does example length affect the benefits of CuBERT? We compare finetuning performance for different example lengths on the classification tasks (Section 4.6).

5. How does CuBERT perform on complex tasks? We implemented and finetuned a model for a multi-headed pointer prediction task, namely, the Variable-Misuse Localization and Repair task (Section 4.7). We compare it to the model from Vasic et al. (2019).

Except for Section 4.6, all the results are presented for sequences of length 512. We give examples of classification instances in Appendix B and include visualizations of attention weights for them.

## 4.3 CONTEXTUAL VS. WORD EMBEDDINGS

The purpose of this analysis is to understand how much pre-trained contextual embeddings help, compared to word embeddings. For each classification task, we trained BiLSTM models starting with each of our baseline Word2Vec embeddings, namely, continuous bag of words (CBOW) and Skipgram trained with negative sampling or hierarchical softmax. In all the models, the Word2Vec embeddings can be refined during training. Within the first 100 epochs, the performance of the BiLSTM models stopped improving. The best model weights per task were selected by finding the minimum validation loss on the corresponding dataset (Table 1) over the first 100 epochs. On the CuBERT side, we finetuned the pre-trained model for 20 epochs, with similar model selection.

The resulting test-split accuracies are shown in Table 2. CuBERT consistently outperforms BiLSTM (with the best task-wise Word2Vec configuration) on all tasks, by a margin of 2.9–22%. Thus, the pre-trained contextual embedding provides superior results even with a smaller budget of 20 epochs, compared to the 100 epochs used for BiLSTMs. The Exception Type classification task is an interesting case since it has an order of magnitude less training data than the other tasks (see

| Best of # Epochs | Train Fraction | Misuse | Operator | Operand | Docstring | Exception |
|---|---|---|---|---|---|---|
| **2** | 100% | 90.09% | 85.15% | 88.67% | 95.81% | 52.38% |
| | 66% | 89.52% | 83.26% | 88.66% | 95.17% | 34.70% |
| | 33% | 88.64% | 82.28% | 87.45% | 95.29% | 26.87% |
| **10** | 100% | 92.73% | 88.43% | 88.67% | 95.81% | 62.55% |
| | 66% | 92.06% | 87.06% | 90.39% | 95.64% | 64.59% |
| | 33% | 91.23% | 84.44% | 87.45% | 95.48% | 54.22% |
| **20** | 100% | 94.61% | 90.24% | 92.56% | 96.85% | 71.74% |
| | 66% | 94.19% | 89.36% | 92.01% | 96.17% | 70.11% |
| | 33% | 93.54% | 87.67% | 91.30% | 96.37% | 67.72% |

Table 3: Effects of reducing training-split size on finetuning performance on the classification tasks.

Table 1). The difference between the performance of BiLSTM and CuBERT is the highest for this task. Thus, finetuning is of much value for tasks with limited labeled training data.

We analyzed the performance of CuBERT with the reduced finetuning budget of only 2 and 10 epochs (see Table 2). Except for the Operand task, CuBERT outperforms BiLSTM within 2 finetuning epochs. On the Operand task, the performance difference between CuBERT with 2 or 10 finetuning epochs and BiLSTM is about 1%. For the rest of the tasks, CuBERT with only 2 finetuning epochs outperforms BiLSTM (with the best task-wise Word2Vec configuration) by a margin of 0.7–12%. This shows that CuBERT can reach accuracies that are comparable to or better than those of BiLSTMs trained with Word2Vec embeddings within only a few epochs.

We also trained the BiLSTM models from scratch, that is, without using the Word2Vec embeddings. The results are shown in the first row of Table 2. Compared to those, the use of Word2Vec embeddings performs better by a margin of 1.5–10.5%. Though no single Word2Vec configuration is the best, CBOW trained with negative sampling gives the most consistent results overall.

## 4.4 IS TRANSFORMER ALL YOU NEED?

One may wonder if CuBERT's promising results derive more from using a Transformer-based model for its classification tasks, and less from the actual, unsupervised pre-training. Here we compare our results on the classification tasks to a Transformer-based model trained from scratch, i.e., without the benefit of a pre-trained embedding. All the models were trained for 100 epochs during which their performance stopped improving. We selected the best model per task using least validation loss. As seen from the last row of Table 2, the performance of CuBERT is substantially higher than the Transformer models trained from scratch. We therefore conclude that pre-training is crucial to CuBERT's success.

## 4.5 THE EFFECTS OF LITTLE SUPERVISION

The big draw of unsupervised pre-training followed by finetuning is that some tasks have small labeled datasets. We study here how CuBERT fares when the size of its training split is reduced. We sampled uniformly the training split of ETH Py150 down to $2/3$rds and $1/3$rd of its original size, and produced training datasets for each of the classification tasks from each sub-split. We then finetuned the pre-trained CuBERT model with each of the 3 different training splits. Validation and testing were done with the same original datasets. Table 3 shows the results.

The Function Docstring task seems robust to the reduction of the training dataset, both early and late in the finetuning process (that is, within 2 vs. 20 epochs), whereas the Exception Classification task is heavily impacted by the dataset reduction, given that it has relatively few training examples to begin with. Interestingly enough, for some tasks, even finetuning for only 2 epochs and only using a third of the training data outperforms the baselines. For example, for both Variable Misuse and Function Docstring, CuBERT at 2 epochs and $1/3$rd training data outperforms the BiLSTM with Word2Vec and the Transformer baselines.

| Length | Misuse | Operator | Operand | Docstring | Exception |
|--------|--------|----------|---------|-----------|-----------|
| 128 | 85.89% | 77.92% | 77.17% | 97.10% | 55.95% |
| 256 | 92.69% | 86.52% | 87.26% | 97.08% | 65.38% |
| 512 | 94.61% | 90.24% | 92.56% | 96.85% | 71.74% |

Table 4: Best out of 20 epochs of finetuning, for three example lengths, on the classification tasks.

| Model | Setting | True Positive | Classification Accuracy | Localization Accuracy | Loc+Repair Accuracy |
|-------|---------|---------------|-------------------------|-----------------------|---------------------|
| **LSTM+pointer** | 100 epochs | 81.63% | 78.76% | 63.83% | 56.37% |
| **CuBERT+pointer** | 2 epochs | **97.18%** | 89.37% | 79.05% | 75.84% |
| | 10 epochs | 94.94% | 93.05% | 88.52% | 85.91% |
| | 20 epochs | 96.83% | **94.85%** | **91.11%** | **89.35%** |

Table 5: Comparison of the finetuned CuBERT+pointer model and the LSTM+pointer model from Vasic et al. (2019) on the variable misuse localization and repair task.

## 4.6 THE EFFECTS OF REDUCING CONTEXT

Context size is especially useful in code tasks, given that some relevant information may lie many "sentences" away from its locus of interest. Here we study how reducing the context length (i.e., the length of the examples used to pre-train and finetune) affects performance. We produce data with shorter example lengths by following the standard BERT mechanism. Table 4 shows the results.

Although context seems to be important to most tasks, the Function Docstring task seems to improve with reduced context. This may be because the task primarily depends on comparison between the docstring and the function signature, and including more context dilutes the model's focus.

For comparison, we also evaluated the BiLSTM model on sequences of length 128 and 256 for the Variable Misuse task. We obtained accuracies of 71.34% and 73.63% respectively, which are lower than the best BiLSTM accuracy on sequence length 512 and also lower than the accuracies of CuBERT for the corresponding lengths (see Table 4).

## 4.7 EVALUATION ON A MULTI-HEADED POINTER PREDICTION TASK

We now discuss the results of finetuning CuBERT to predict the localization and repair pointers for the variable misuse task. For this task, we implement the multi-headed pointer model from Vasic et al. (2019) on top of CuBERT. The baseline consists of the same pointer model on a unidirectional LSTM as used in Vasic et al. (2019). We refer to these respectively as CuBERT+pointer and LSTM+pointer models, respectively. Due to limitations of space, we omit the details of the pointer model and refer the reader to the above paper. As reported in Section 4 of Vasic et al. (2019), to enable comparison with an enumerative approach, the evaluation was performed only on 12K test examples. In comparison, we report the numbers on all 430K test examples (Table 1) for both the models.

Similar to other tasks, we trained the baseline model for 100 epochs and finetuned CuBERT for up to 20 epochs. Table 5 gives the results along the same metrics as Vasic et al. (2019). The metrics are defined as follows: 1) True Positive is the percentage of bug-free functions classified as bug-free. 2) Classification Accuracy is the percentage of correctly classified examples (between bug-free and buggy). 3) Localization Accuracy is the percentage of buggy examples for which the localization pointer correctly identifies the bug location. 4) Localization+Repair Accuracy is the precentage of buggy examples for which both the localization and repair pointers make correct predictions. As seen from Table 5, the CuBERT+pointer model outperforms the LSTM+pointer model consistently across all the metrics, and even within 2 and 10 epochs.

## 5 CONCLUSIONS AND FUTURE WORK

We present the first attempt at pre-trained contextual embedding of source code by training a BERT model, called CuBERT, which we finetuned on five classification tasks and compared against BiL-STM with Word2Vec embeddings and Transformer models. As a more challenging task, we also evaluated CuBERT on a multi-headed pointer prediction task. CuBERT outperformed the baseline models consistently. We evaluated CuBERT with less data and fewer epochs, highlighting the benefits of pre-training on a massive, unsupervised code corpus. We see this as a promising step towards source-code understanding, and plan to explore its utility on other programming languages and tasks. We use a tokenized representation of source code and leave it to the underlying Transformer model to infer any structural interactions between the input tokens through its self-attention mechanism. However, the literature on deep learning for source code also demonstrates the utility of encoding explicit structural information such as data-flow information (Allamanis et al., 2018). The Transformer model has been extended to incorporate explicitly provided structural information (Shaw et al., 2018). Using such relation-aware Transformers for pre-training source code representations will be an important direction to explore in future.

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

| Exception Type | Test | Validation | Train 100% | Train 66% | Train 33% |
|---|---|---|---|---|---|
| ValueError | 2324 | 477 | 4058 | 2715 | 1344 |
| KeyError | 2240 | 453 | 4009 | 2566 | 1271 |
| AttributeError | 1657 | 311 | 2895 | 1896 | 876 |
| TypeError | 913 | 187 | 1747 | 1175 | 564 |
| OSError | 891 | 164 | 1641 | 1106 | 543 |
| IOError | 865 | 168 | 1560 | 1046 | 560 |
| ImportError | 776 | 202 | 1372 | 935 | 471 |
| IndexError | 694 | 153 | 1197 | 813 | 408 |
| DoesNotExist | 6 | 2 | 3 | 0 | 0 |
| KeyboardInterrupr | 287 | 67 | 590 | 408 | 223 |
| StopIteration | 307 | 69 | 488 | 302 | 155 |
| AssertionError | 177 | 32 | 397 | 276 | 158 |
| SystemExit | 139 | 23 | 264 | 173 | 101 |
| RuntimeError | 128 | 36 | 299 | 203 | 104 |
| HTTPError | 59 | 13 | 119 | 80 | 35 |
| UnicodeDecodeError | 151 | 24 | 251 | 173 | 82 |
| NotImplementedError | 127 | 27 | 222 | 136 | 52 |
| ValidationError | 95 | 15 | 172 | 121 | 58 |
| ObjectDoesNotExist | 105 | 17 | 213 | 142 | 64 |
| NameError | 95 | 19 | 197 | 124 | 56 |

Table 6: Example counts per class for the Exception Type task, broken down into the dataset splits. We show separately the 100% train dataset, as well as its 33% and 66% subsamples used in the ablations.

# A  DATA PREPARATION FOR FINETUNING TASKS

## A.1  LABEL FREQUENCIES

All four of our binary-classification finetuning tasks had an equal number of buggy and bug-free examples. The Exception task, which is a multi-class classification task, had a different number of examples per class (i.e., exception types). We show the breakdown of example counts per label for our finetuning dataset splits in Table 6.

## A.2  FINETUNING TASK DATASETS

In this section, we describe in detail how we produced our finetuning datasets (Section 3.4).

A common primitive in all our data generation is splitting a Python module into functions. We do this by parsing the Python file and identifying function definitions in the Abstract Syntax Tree that have no other function definition between themselves and the root of the tree. The resulting functions include functions defined at module scope, but also methods of classes and subclasses. Not included are functions defined within other function and method bodies, or methods of classes that are, themselves, defined within other function or methods bodies.

We do not filter functions by length, although task-specific data generation may filter out some functions (see below). When generating examples for a fixed-length pre-training or finetuning model, we prune all examples to the maximum target sequence length (in this paper we consider 128, 256, and 512 subtokenized sequence lengths). Note that if a synthetically generated buggy/bug-free example pair differs only at a location beyond the target length (say on the 600-th subtoken), we still retain both examples. For instance, for the Variable Misuse Localization and Repair task, we retain both buggy and bug-free examples, even if the error and/or repair locations lie beyond the end of the maximum target length.

| | **Commutative** | **Non-Commutative** |
|---|---|---|
| **Arithmetic** | +, * | -, /, % |
| **Comparison** | ==, !=, is, is not | <, <=, >, >= |
| **Membership** | | in, not in |
| **Boolean** | and, or | |

Table 7: Binary operators.

### A.2.1 REPRODUCIBLE DATA GENERATION

We make pseudorandom choices at various stages in finetuning data generation. It was important to design a pseudorandomness mechanism that gave (a) reproducible data generation, (b) non-deterministic choices drawn from the uniform distribution, and (c) order independence. Order independence is important because our data generation is done in a distributed fashion (using Apache Beam), so different pseudorandom number generator state machines are used by each distributed worker.

More specifically, pseudorandomness is computed based on an experiment-wide seed, but is independent of the order in which examples are generated. Specifically, to make a pseudorandom choice about a function, we hash (using MD5) the seed and the function data (its source code and metadata about its provenance), and use the resulting hash as a uniform pseudorandom value from the function, for whatever needs the data generator has (e.g., in choosing one of multiple choices). In that way, the same function will always result in the same choices given a seed, regardless of the order in which each function is processed, resulting in reproducible dataset generation.

To choose among multiple choices, we hash the function's pseudorandom value along with all choices (sorted in a canonical order) and use the digest to compute an index within the list of choice. Note that given two choices over different candidates but for the same function, independent decisions will be drawn. We also use such order-independent pseudorandomness when subsampling datasets (e.g., to generate the validation datasets). In those cases, we hash a sample with the seed, as above, and turn the resulting digest into a pseudorandom number in $[0, 1]$, which can be used to decide given a target sampling rate.

### A.2.2 VARIABLE MISUSE CLASSIFICATION

A variable use is any mention of a variable in a load scope. This includes a variable that appears in the right-hand side of an assignment, or a field dereference. We regard as *defined* all variables mentioned either in the formal arguments of a function definition, or on the right-hand side of an assignment. We do not include in our defined variables those declared in module scope (i.e., globals).

To decide whether to generate examples from a function, we parse it, and collect all variable-use locations, and all define variables, as described above. We discard the function if it has no variable uses, or if it defines fewer than two variables; if there is only one variable defined, the problem of detecting variable misuse is moot. For any function that we do not discard, we generate a buggy and a bug-free example, as described next.

To generate a buggy example from a function, we choose one variable use pseudorandomly (see above how multiple-choice decisions are done), and replace its current occupant with a different pseudorandomly-chosen variable defined in the function (with a separate multiple-choice decision).

### A.2.3 WRONG BINARY OPERATOR

This task considers both commutative and non-commutative binary operators (unlike the Swapped-Argument Classification task). See Table 7 for the full list, and note that we have excluded relatively infrequent operators, e.g., the Python integer division operator //.

If a function has no binary operators, it is discarded. Otherwise, it is used to generate a bug-free example, and a single buggy example as follows: one of the operators is chosen pseudorandomly (as described above), and a different operator chosen to replace it in the same row of the Table 7. So, for instance, a buggy example would only swap == with is, but not with not in, which would not type-check if we performed static type inference on Python.

We take appropriate care to ensure the code parses after a bug is introduced. For instance, if we swap the operator in the expression `1==2` with `is`, we ensure that there is space between the tokens (i.e., `1 is 2` rather than the incorrect `1is2`), even though it was not needed before.

### A.2.4 SWAPPED OPERAND

Since this task targets swapping the arguments of binary operators, we only consider non-commutative operators from Table 7.

Functions without eligible operators are discarded, and the choice of the operator to mutate in a function, as well as the choice of buggy operator to use, are done as above, but limiting choices only on non-commutative operators.

To avoid complications due to format changes, we only consider expressions that fit in a single line (in contrast to the Wrong Binary Operator Classification task). We also do not consider expressions that look the same after swapping (e.g., `a - a`).

### A.2.5 FUNCTION-DOCSTRING MISMATCH

In Python, a function docstring is a string literal that directly follows the function signature and before the main function body. Whereas in other common programming languages, the function documentation is a comment, in Python it is an actual, semantically meaningful string literal.

We discard functions that have no docstring from this dataset, or functions that have an empty docstring. We split the rest into the function definition without the docstring, and the docstring summary (i.e., the first line of text from its docstring), discarding the rest of the docstring.

We create bug-free examples by pairing a function with its own docstring summary.

To create buggy examples, we pair every function with another function's docstring summary, according to a global pseudorandom permutation of all functions: for all $i$, we combine the $i$-th function (without its docstring) with the $P_i$-th function's docstring summary, where $P$ is a pseudorandom permutation, under a given seed. We discard pairings in which $i == P[i]$, but for the seeds we chose, no such pathological permuted pairings occurred.

### A.2.6 EXCEPTION TYPE

Note that, unlike all other tasks, this task has no notion of buggy or bug-free examples.

We discard functions that do not have any `except` clauses in them.

For the rest, we collect all locations holding exception types within `except` clauses, and choose one of those locations to query the model for classification. Note that a single `except` clause may hold a comma-separated list of exception types, and the same type may appear in multiple locations within a function. Once a location is chosen, we replace it with a special `__HOLE__` token, and create a classification example that pairs the function (with the masked exception location) with the true label (the removed exception type).

The count of examples per exception type can be found in Table 6.

### A.2.7 VARIABLE MISUSE LOCALIZATION AND REPAIR

The dataset for this task is identical to that for the Variable Misuse Classification task (Section A.2.2). However, unlike the classification task, examples contain more features relevant to localization and repair. Specifically, in addition to the token sequence describing the program, we also extract a number of boolean input masks:

- A *candidates* mask, which marks as True all tokens holding a variable, which can therefore be either the location of a bug, or the location of a repair. The first position is always a candidate, since it may be used to indicate a bug-free program.

- A *targets* mask, which marks as True all tokens holding the correct variable, for buggy examples. Note that the correct variable may appear in multiple locations in a function,

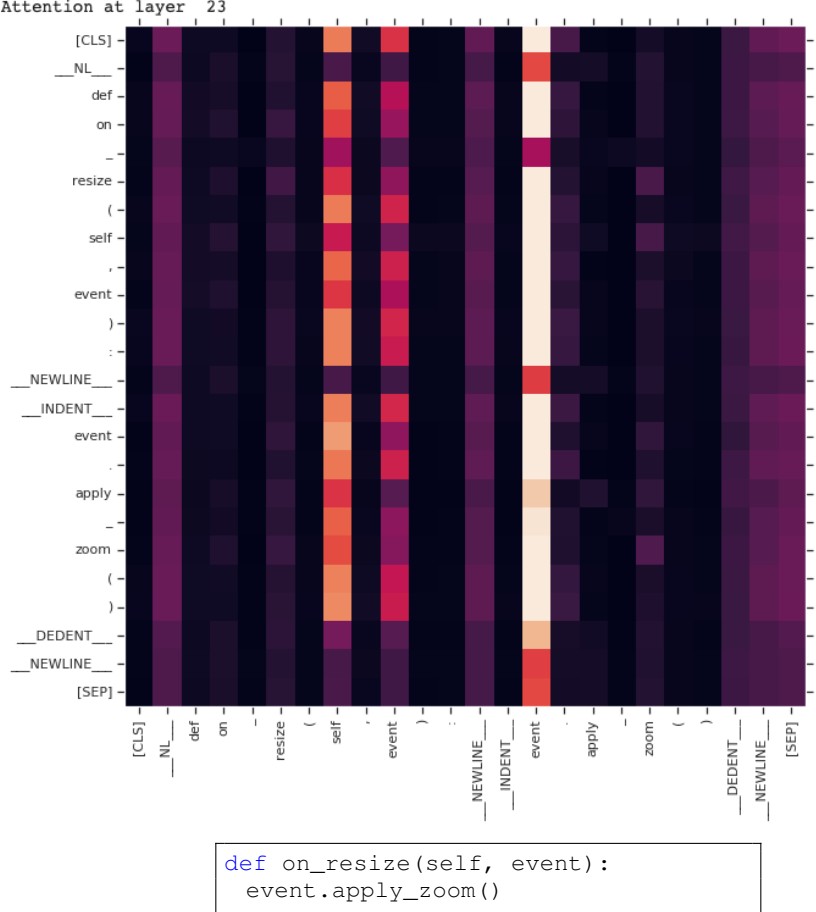

```
def on_resize(self, event):
    event.apply_zoom()
```

Figure 1: Variable Misuse Example. In the code snippet, 'event.apply_zoom' should actually be 'self.apply_zoom'. The CuBERT variable-misuse model correctly predicts that the code has an error. As seen from the attention map, the query tokens are attending to the second occurrence of the 'event' token in the snippet, which corresponds to the incorrect variable usage.

therefore this mask may have multiple True positions. Bug-free examples have an all-False targets mask.

- An *error-location* mask, which marks as True the location where the bug occurs (for buggy examples) or the first location (for bug-free examples).

All the masks mark as True some of the locations that hold variables. Because many variables are subtokenized into multiple tokens, if a variable is to be marked as True in the corresponding mask, we only mark as True its first subtoken, keeping trailing subtokens as False.

## B   ATTENTION VISUALIZATIONS

In this section, we provide sample code snippets used to test the different classification tasks. Further, Figures 1–5 show visualizations of the attention matrix of the last layer of the finetuned CuBERT model (Coenen et al., 2019) for the code snippets. In the visualization, the Y-axis shows the query tokens and X-axis shows the tokens being attended to. The attention weight between a pair of tokens is the maximum of the weights assigned by the multi-head attention mechanism. The color changes from dark to light as weight changes from 0 to 1.

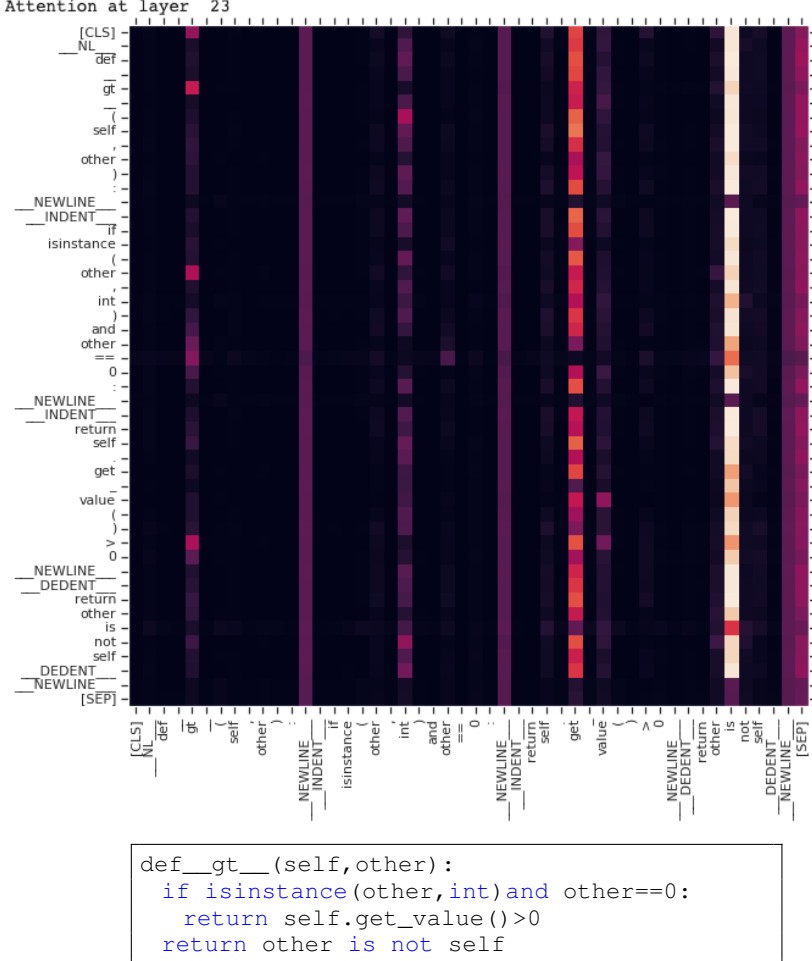

Figure 2: Wrong Operator Example. In this code snippet, 'other is not self' should actually be 'other < self'. The CuBERT wrong-binary-operator model correctly predicts that the code snippet has an error. As seen from the attention map, the query tokens are all attending to the incorrect operator 'is'.

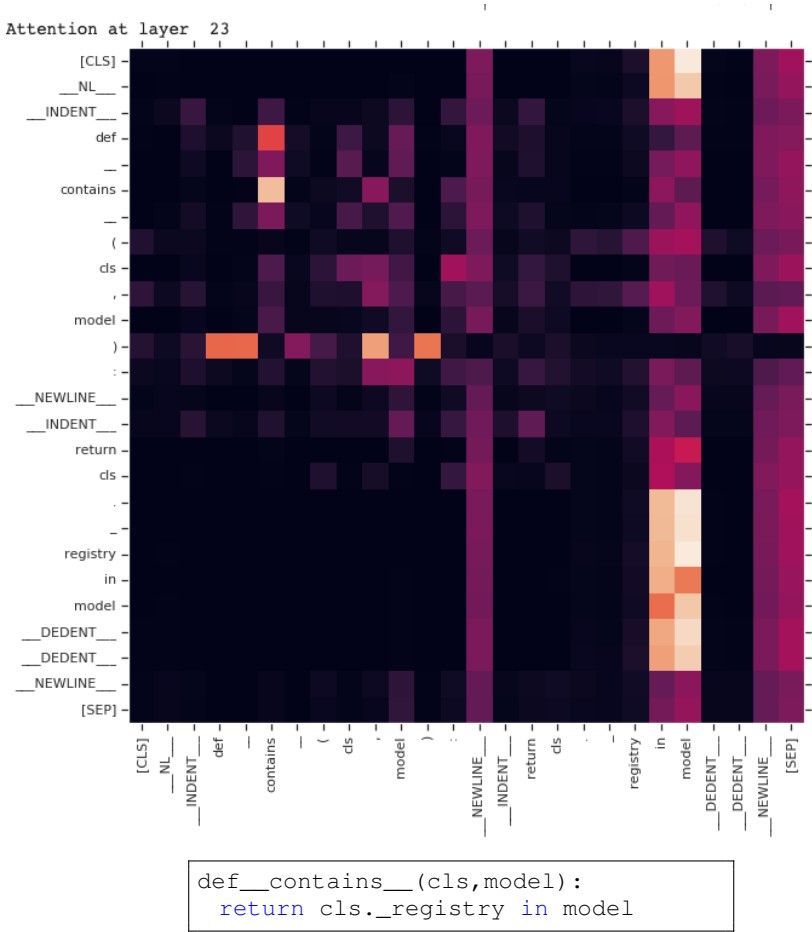

Figure 3: Swapped Operand Example. In this code snippet, the return statement should be 'model in cls._registry'. The swapped-operand model correctly predicts that the code snippet has an error. The query tokens are paying substantial attention to 'in' and the second occurrence of 'model' in the snippet.

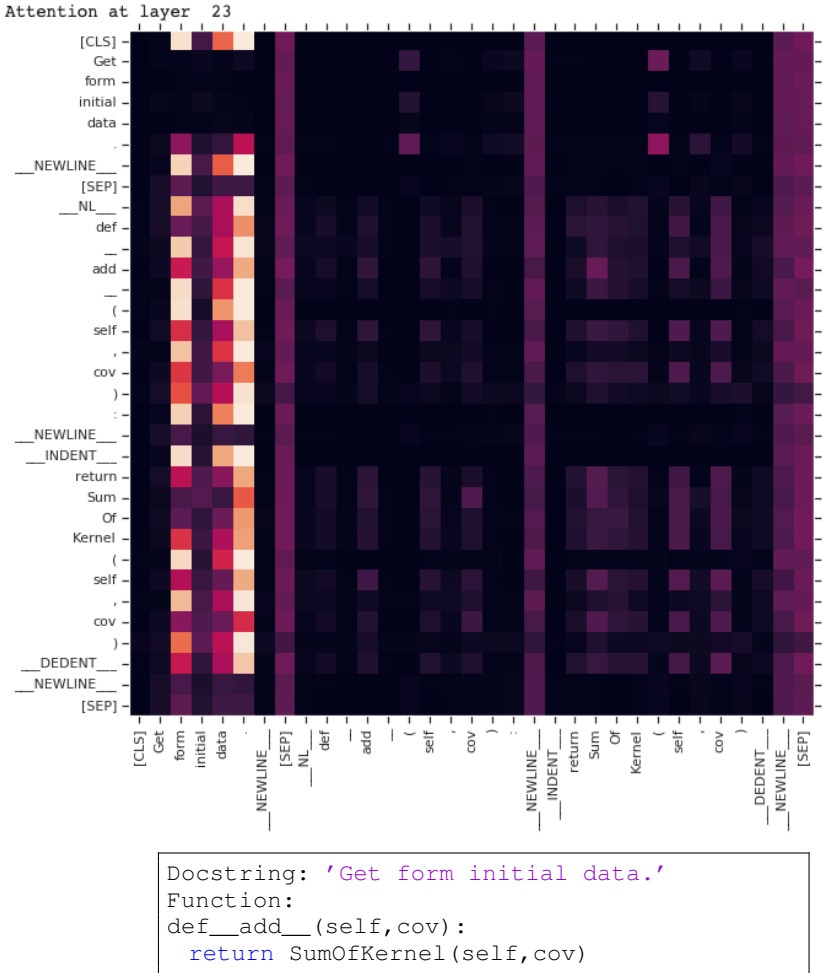

Figure 4: Function Docstring Example. The CuBERT function-docstring model correctly predicts that the docstring is wrong for this code snippet. Note that most of the query tokens are attending to the tokens in the docstring.

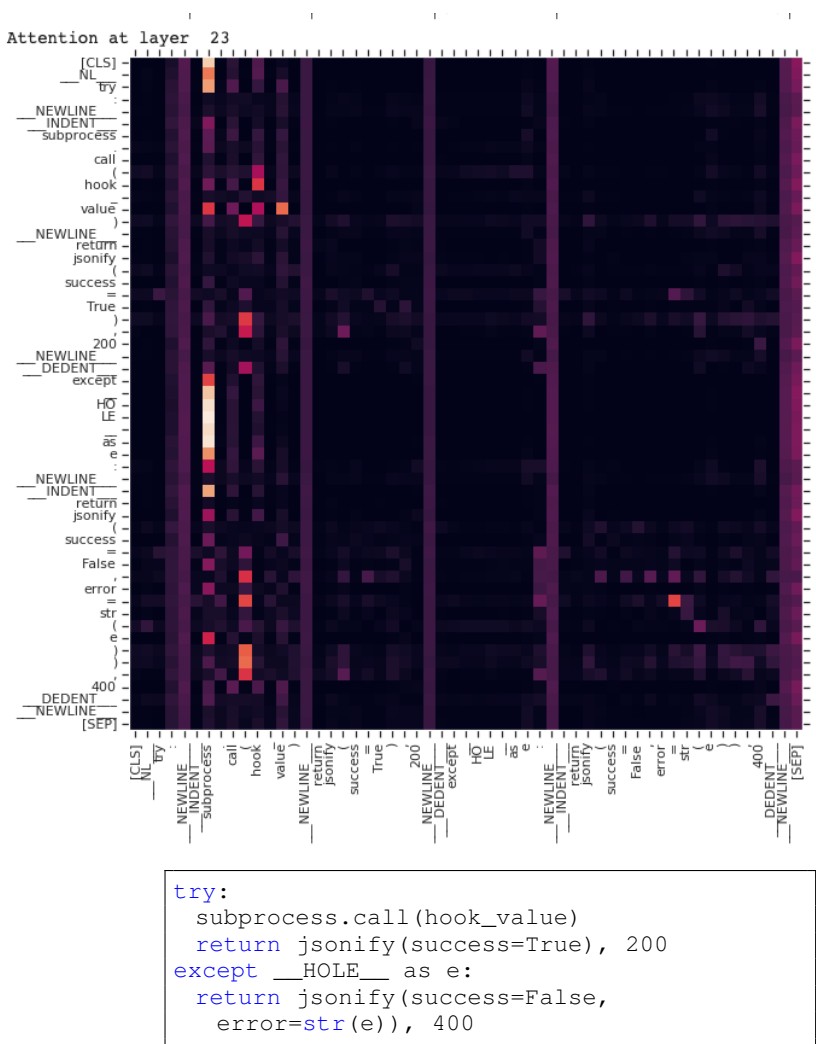

```
try:
    subprocess.call(hook_value)
    return jsonify(success=True), 200
except __HOLE__ as e:
    return jsonify(success=False,
        error=str(e)), 400
```

Figure 5: Exception Classification Example. For this code snippet, the CuBERT exception-classification model correctly predicts '␣␣HOLE␣␣' as 'OSError'. The model's attention matrix also shows that '␣␣HOLE␣␣' is attending to 'subprocess', which is indicative of an OS-related error.

