# OpenReview forum: "Pre-trained Contextual Embedding of Source Code"
_ICLR.cc/2020/Conference — Reject_

### Official Review · AnonReviewer2 · 2019-10-08
**Official Blind Review #2**

**Rating:** 6

**Review:**

This paper describes a BERT-based pre-training for source code related task. By pre-training on BERT-like models on source code and finetuning on a set of 5 tasks, the authors show good performance improvements over non-pretrained models. The authors make a series of ablation studies showing that pre-training is indeed useful.

Overall, I find this work relevant and interesting, albeit somewhat unsurprising. Nevertheless, I see no reason to reject this paper. To make my "weak accept" to a "strong accept" I would like to see experiments on more tasks, preferably more complex tasks. For example, such tasks could include (a) variable naming (b) method naming (c) docstring prediction/summarization (d) language modeling/autocompletion. I believe it's unclear to the reader if pre-training is also helpful for any of those tasks too and adding such experiments would significantly strengthen the paper.

Some clarifications comments/questions to the authors:

* I would insist that the authors rename the "Variable Misuse" task to "Variable Misuse Localization". To my understanding the current model points to the misused variable (if any), but does not attempt to suggest a fix. This tackles only a part of the task discussed in Vasic et al. (2019), Allamanis et al (2018) and this might confuse readers who want to compare with those works.

* For the Function-Docstring Mismatch task (Section 3.4):
    * It's unclear to me which dataset is used. Is it the Py150 dataset or the Barone & Sennrich (2017)?
    * I believe that the citations [a], [b] would be appropriate here.

* Overall, for the all the tasks except from "Exception Type", there is a replicability issue: Since the authors manually mutate the code (e.g. introduce a variable misuse, swap an operand), for anyone to compare directly, they would need access to the mutated samples. I would strongly encourage the authors to provide more details on how they create mutated samples and (eventually) the source code that achieves that.

* For the Variable Misuse, Wrong Binary Operator, Swapped Operand tasks. There are a few things that need to be clarified:
   * How long is each code snippet? One would expect that the longer the code snippet the harder the task. Do the authors pass a whole function?
   * What is the proportion of positive/negative examples in each task?



[a] Cambronero, Jose, et al. "When Deep Learning Met Code Search." arXiv preprint arXiv:1905.03813 (2019).
[b] Louis, Annie, et al. "Deep learning to detect redundant method comments." arXiv preprint arXiv:1806.04616 (2018).

**Experience Assessment:**

I have published in this field for several years.

**Review Assessment: Checking Correctness Of Derivations And Theory:**

N/A

**Review Assessment: Checking Correctness Of Experiments:**

I assessed the sensibility of the experiments.

**Review Assessment: Thoroughness In Paper Reading:**

I read the paper at least twice and used my best judgement in assessing the paper.

---

> ### Author Response · Authors · 2019-11-13
> **Response to Review #2**
>
> We thank the reviewer for the helpful comments and suggestions.
>
> >> Addition of more complex task
>
> We have now added a more complex task (Section 4.7), that of joint classification, localization and repair of variable misuse errors as proposed in Vasic et al. (2019). This requires learning two pointers for localization and repair of variable misuse bugs, and is an extension of the variable misuse classification task we have already considered.
>
> We had considered the variable and method naming tasks as candidate finetuning tasks. The masked language modeling (MLM) pre-training task works by masking/replacing tokens (Section 3.5) and training the network to predict them. Variable and method naming tasks would be very similar to MLM (wherein we mask the names and ask the network to predict the masked tokens) and hence, we decided not to include them in this submission.
>
> Since CuBERT produces only a pre-trained encoder, the docstring prediction and language modeling/autocomplete tasks would require learning a decoder from scratch. A recent work on transfer learning (“Exploring the Limits of Transfer Learning with a Unified Text-to-Text Transformer”, https://arxiv.org/abs/1910.10683) recasts the BERT pre-training objective into a text-to-text setting, wherein both Transformer encoder and decoder are pre-trained. We consider this reformulation of BERT to be a great future avenue to enable direct finetuning for generative tasks like docstring prediction and autocompletion.
>
> >> Renaming the Variable Misuse task
>
> The task we had described was a classification task where the model needs to identify if any of the variables in a function body is misused. To avoid any misunderstanding, we have now renamed it to “Variable Misuse Classification”. To also match the full task from Vasic et al. (2019), we now also have the “Variable Misuse Localization and Repair” task (Section 4.7).
>
> >> Dataset for the Function-Docstring Mismatch task and related references
>
> The Py150 dataset is used in this task (and all other fine-tuning tasks). We have updated the writeup to make this clear. Thank you for the references, we have discussed them in the writeup now.
>
> >> Details on reproducibility
>
> We have added an appendix (Appendix A) with the details of the dataset generation for the finetuning tasks, including a discussion of the careful use of pseudorandomness to ensure reproducible dataset generation. In addition, we plan to release the datasets for public use.
>
> >> Details on data generation and proportion of positive/negative examples
>
> We have included these details in Appendix A in the revised version.

---

> > ### Comment · AnonReviewer2 · 2019-11-14
> > **Thanks**
> >
> > Thank for your responses. I find the comments above satisfactory and the additional experiments helpful.
> >
> > On the concerns of Reviewer #1, I understand that comparison with past work is not possible at this time (which is annoying) but due to the varied nature of the data that previous work has used. I'd argue that publishing the data used in this paper _could_ be a step towards achieving this in the community.  Because of this, I would like to argue for the acceptance of this work on the condition of publishing all the relevant (real and synthetic) data.

---

> > > ### Comment · AnonReviewer1 · 2019-11-14
> > > **On datasets**
> > >
> > > While I also like the idea that the data should be public to help further work on top of the paper,  I do not believe it is right to demand it from the authors as a condition for accepting the paper.
> > >
> > > What is important is that not all their results are independent from the rest of the world, because this would limit the ability of the reader to compare it with future or past techniques (also will make it worse for everyone after them to publish without ignoring their work).

---

> > > > ### Comment · AnonReviewer2 · 2019-11-14
> > > > **Dataset**
> > > >
> > > > Given that the modeling contributions of this work are relatively weak (CuBERT is a BERT architecture with relatively small changes) I am arguing for a "weak accept". A public dataset augments the existing contributions. So, to clarify my original response: Should the authors wish to keep the data private, my recommendation is still a "weak accept", but I won't fight for this work to get accepted and I can see the arguments for a "weak reject". If the data will be made public, then this makes this work more valuable and I am willing to argue for it to be accepted to the AC and the other reviewers.
> > > >
> > > > Datasets are valuable contributions. Papers need to be rewarded for the data collection/generation efforts independently of other methodological contributions to the field. I believe that ICLR should welcome dataset-based papers on important problems.
> > > >
> > > > As R1 suggests, baselines on problems are important. But reimplementing them every time on new datasets (for different programming languages) isn't a realistic way for this field to progress. I suspect that the authors of this work have stumbled upon this problem (hence the lack of baselines/comparisons with prior work, e.g. Pradel et al. 2018, Allamanis et al. 2015, Allamanis et al. 2018, Cambronero et al. 2019, Raychev et al. 2015, etc). I don't think that the authors of this work should be penalized for the lack of good public datasets/code on the problems they study for a single programming language (Python in their case).

---

> > > > > ### Author Response · Authors · 2019-11-14
> > > > > **Dataset Availability**
> > > > >
> > > > > We agree that public benchmarks are important for this community. It is true that we had to expend considerable effort to prepare datasets for this work and wish to contribute the outcome to the community. Our plan is to publish 1) the list of files (and versions) in our deduplicated pre-training dataset -- the actual files are already publicly available in GitHub or BigQuery's public GitHub datastore; 2) our entire finetuning datasets, 3) our pre-trained/finetuned models, 4) associated utilities, and 5) relevant vocabularies.

---

> > > > > > ### Comment · Area_Chair1 · 2019-11-15
> > > > > > **Re: Dataset Availability**
> > > > > >
> > > > > > It would be great if the datasets and code can be shared so that readers can reproduce the results in the paper.

---

### Official Review · AnonReviewer1 · 2019-10-23
**Official Blind Review #1**

**Rating:** 3

**Review:**

The paper proposes a transformer based approach to address a number of problems for machine learning from code. Then, the paper adds BERT-based pretraining to significantly improve on the results. The paper is nicely written and easy to follow, but with relatively thin contributions besides the large amount of experiments.

Initially, I was quite positive on the entire paper, but as I read it in more details, I got less convinced that this is something I want to see at ICLR. First, there are no good baselines. BiLSTM gets quite high accuracy on most of the tasks, which is unexpected because most prior works show that the tasks benefit from some semantic understanding of code. I cannot relate any of the numbers with a previous work. Right now, I even have reduced confidence that the authors do not have bugs in the tasks or the reported numbers. Then, for the Operator and Operand tasks, BiLSTM is also doing impressively well (these tasks were done differently in prior works). Interestingly, things get reversed on the last two tasks. Given that most of the experiments are not the same as in any previous paper, I would strongly appreciate if much more details are given in the appendix. In fact, the appendix right now does not have much useful information besides praising how good job the attention is doing. What would be needed is information on how many samples were generated, how large were they, was any data thrown out? Table 1 is a good start, but it actually raises questions. You split the Python dataset into separate functions like Vasic et al and the number of functions 2x higher (counting the artificial samples, I guess), did you put a limit on how large functions you consider? 250 tokens was the limit of Vasic et al. To which of the tasks in the Vasic et al paper can I relate? Is the BiLSTM on the level of that work or it is substantially worse or better? Also, are the results coming from the paper SOTA or uncomparable to other works?

Five tasks are evaluated, which is impressive. This is one reason I want to see the results published. The problem is also quite important. The experiments that show the effect on reduced labelled data are quite important and interesting - in fact, for many tasks, we can start curating datasets and having model working on small data will be crucial. However, I think the paper needs more work before it is something to present, cite or build upon.


**Experience Assessment:**

I have published one or two papers in this area.

**Review Assessment: Checking Correctness Of Derivations And Theory:**

N/A

**Review Assessment: Checking Correctness Of Experiments:**

I carefully checked the experiments.

**Review Assessment: Thoroughness In Paper Reading:**

I read the paper at least twice and used my best judgement in assessing the paper.

---

> ### Author Response · Authors · 2019-11-13
> **Response to Review #1**
>
> We thank the reviewer for the helpful comments and suggestions.
>
> >> Lack of good baselines
>
> Though we consider a variety of tasks from the literature, the associated datasets came from different languages and varied sources. Due to the importance of keeping the pre-training corpus distinct from the finetuning corpus, we decided to set up all our finetuning tasks on the ETH Py150 corpus and constructed strong and suitable baselines. As baselines, we use multi-layered BiLSTMs and Transformers, both of which are widely-used architectures for sequential data. Since we study the transferability of pre-trained contextual embeddings, we have also trained and used 4 variants of (non-contextual) Word2Vec word embeddings. To our knowledge, these Word2Vec embeddings also constitute the only word embeddings trained for source code at a massive scale. As a result, we compare with not only strong architectures but with also pre-trained word embeddings of source code.
>
> >> Justification for high accuracy of BiLSTM models
>
> There is certainly work in the literature that shows that results can be improved by incorporating additional annotations in the program representations. However, there is also a large body of work that shows impressive results with only tokenized source code representation. In the absence of systematic benchmarking and comparisons, we shy away from claiming either of them to be superior in general. For this work, we have put in extensive efforts to ensure that the baseline models are not at a disadvantage when compared to CuBERT. In particular, we have purposefully chosen bidirectional LSTMs since the Transformer encoder underlying CuBERT uses bidirectional context. Another potential reason for our BiLSTM models doing well is that many existing works in the literature use token-level vocabularies whose size is capped. This can result in out-of-vocabulary words and hampers the learning. We use a subword vocabulary (the same one that we use for CuBERT) which avoids this. Further, we use well-trained Word2Vec embeddings in our BiLSTM models.
>
> >> Variation in the performance of the BiLSTM models
>
> Note that compared to the other classification tasks, the Exception Type classification task is a multi-class problem with 20 classes and has a much smaller dataset (21K training examples). So the BiLSTMs do not perform that well on this task. In general, the model performance is subject to idiosyncrasies of the problem and dataset.
>
> >> Comparison to previous work
>
> For reasons discussed above (under “Lack of good baselines”), we had to design baselines ourselves. Nevertheless, we now present an additional pointer prediction task (Variable Misuse Localization and Repair), in which we compare CuBERT against the model proposed in Vasic et al. (2019) on our dataset.
>
> >> Appendix explaining details of design of finetuning tasks
>
> Thank you for the suggestion to add more details about the design of the finetuning tasks in the appendix. We have added an appendix (Appendix A) with the requisite details and additional discussion.
>
> >> Comparison with Vasic et al. (2019)
>
> To facilitate direct comparison between Vasic et al. (2019) and our work, we have now added the joint classification, localization and repair task proposed in Vasic et al. (2019). The results are presented in Section 4.7. The results are consistent with the results of our other finetuning tasks, in the sense that CuBERT outperforms the model from Vasic et al. (2019). For fair comparison, the results are obtained on the same dataset for both CuBERT and Vasic et al.’s model (a unidirectional LSTM with two pointers). In their paper, they had done the evaluation only on a random sample of 12,218 test examples (Section 4, Benchmarks, from their paper), whereas we provide results on all of 430K test examples (Table 1, from our paper). We also use the same subword vocabulary for both models, whereas they had used a word-level fixed-size vocabulary.
>
> >> Are the results from the paper SOTA?
>
> Our finetuning tasks are motivated by the tasks in the literature. The reason we do not have one-to-one comparison with existing works is because the existing works use different languages, datasets, tokenizations, etc. (Section 3.4). For example, Pradel & Sen (2018) use limited surrounding context for classification, whereas we do classification at the level of function bodies. Therefore, even though our results are consistently high (>90% accuracy except for the Exception Type task), there are no references that we can appeal to for comparison. However, we now also have results on Variable Misuse Localization and Repair task (Section 4.7) where we evaluate against the model from Vasic et al. (2019) on our dataset, where CuBERT attains the highest accuracies.

---

> > ### Comment · AnonReviewer1 · 2019-11-14
> > **Thanks**
> >
> > Thank you for adding the additional results and discussing the way the dataset was collected. With this, at least one task can now be related to previous work. I know there is limited time, but at least at a later revision, it would be much more clear if the baselines for this task of Vasic et al are included in Table 5 (RNN w/o pretraining as in Vasic et al.).

---

> > > ### Author Response · Authors · 2019-11-14
> > > **About the LSTM+pointer line in Table 5.**
> > >
> > > We would like to clarify that the LSTM+pointer line in Table 5 represents exactly the model from Vasic et al. It's a unidirectional LSTM, with the pointer-prediction layers described in that work. Similarly to Vasic et al., it is trained end-to-end (without pre-training) on the task dataset.
> > >
> > > Can you please confirm if this addresses your comment about this table?

---

> > > > ### Comment · AnonReviewer1 · 2019-11-14
> > > > **yes**
> > > >
> > > > yes

---

### Official Review · AnonReviewer3 · 2019-10-25
**Official Blind Review #3**

**Rating:** 6

**Review:**

# Summary

This paper presents a BERT-inspired pretraining/finetuning setup for source code tasks. It collects a corpus of
unlabeled Python files for BERT pretraining, designs or adopts 5 tasks on established smaller-scale Python corpora, and
adjusts the BERT model to tokenize and encode source code snippets appropriately.

# Strengths

* The idea of applying the pretraining/finetuning paradigm to program analysis tasks makes sense, and has been
  informally attempted by multiple groups in the community in 2019. This is the first high-quality submission to a
  top-tier ML conference I've seen on the subject, though.
* The authors exercised commendable care and diligence in preparing the training data, adopting BERT to source code
  inputs, and ensuring correctness of the experimental setup. I appreciated all the provided details on tokenization
  (Section 3.3), deduplication (Sections 3.1-3.2), and task setup (Section 3.5). This should become a technical standard
  in the community.
* The paper is written clearly and concisely, and is generally a pleasure to read.

# Weaknesses

I have a gripe with the authors' choice to ignore program structure (e.g. abstract syntax trees) or features (e.g.
types) in their program representation. Without this extra information (easily available from a compiler/interpreter
API) the pipeline is not substantially different from the original NLP pipeline of BERT et al. The main program-related
representation insight comes in tokenization (Section 3.3) and the definition of "sentences". To repeat, I appreciate
the effort the authors put in making tokenization appropriate for BERT processing of source code, but this is a drop in
the bucket compared to the all the other program-related features the work is leaving off the table. Programs are not
natural language.
The argument that source code analysis would "pass on the burden ... to downstream tasks" (Page 3) is odd. First, most
downstream tasks of interest occur in the settings where this analysis is already available: IDEs, code review
assistants, linters, etc. Second, one often needs program analysis to even define downstream tasks in the first place --
for example, determining whether function arguments are swapped required detecting a function call and boundaries of its
arguments, thus parsing the program!

This work obtains (and nicely analyzes) impressive results obtained by applying CuBERT. However, it does not put the
results in context with prior work based on structured program representations. Without this, it is difficult to say
whether the improvement comes from pretraining or from the language model simply learning a better "parsed"
representation of an input program from all the unlabeled corpus. If it's the latter, one might argue that supplying the
model with structured program features explicitly might eliminate much of the need for the unlabeled corpus.
I personally think that there will still be a gap between pretraining and finetuning even with structured program
features simply due to the sheer volume of available data, which, as the authors showed, is crucial for good
generalization of Transformer. However, this still needs to be shown empirically.

The "Function-Docstring Mismatch" task, as presented, seems too easy. If the distractors (negative examples) are truly
chosen at random, most of them are going to use obviously different vocabulary from the original function signature (as
Figure 4 demonstrates). A well designed task would somehow bias the sampling toward subtle distractors such as `get` vs.
`set` docstrings, but this seems challenging.
This also explains why the task is not influenced as much by reduction of training data (Table 3).

The Next Sentence Prediction pretraining task, as adapted for CuBERT, seems too difficult, in contrast. If the paired
sentences (i.e. code lines) are chosen at random, the model would lack most of the context required to make a decision
about the logical relationship between them, such as which variables are defined and available in context, which
functionality is being implemented, etc. I wonder, can the authors experiment with pretraining CuBERT only with the
Masked Language Model task? Will it worsen the results substantially or at all?

# Questions

Section 3.2: "similar files according to the same similarity metric..."
What are these metrics?

What is the fraction of positive/negative examples in the constructed finetuning datasets?

What is the motivation for making Variable Misuse and Wrong Operator/Operand into a simple classification tasks instead
of the original (more useful) correction task?


**Experience Assessment:**

I have published one or two papers in this area.

**Review Assessment: Checking Correctness Of Derivations And Theory:**

I carefully checked the derivations and theory.

**Review Assessment: Checking Correctness Of Experiments:**

I carefully checked the experiments.

**Review Assessment: Thoroughness In Paper Reading:**

I read the paper at least twice and used my best judgement in assessing the paper.

---

> ### Author Response · Authors · 2019-11-13
> **Response to Review #3**
>
> We thank the reviewer for the helpful comments and suggestions.
>
> >> Choice to ignore program structure (e.g. abstract syntax trees) or features (e.g. types)
>
> Natural languages are also endowed with structure (e.g., different types of parse trees, phrase structures, etc.). However, the prevailing pre-training methods in NLP such as BERT do not make explicit use of such structure, and still attain state-of-the-art results. The task of learning useful (structural) features is left to the self-attention mechanism of the Transformer model. In this work, we apply the same approach to program-understanding tasks. We recognize that it may be possible to extend CuBERT with explicitly provided structural information using approaches like relation-aware Transformers (see "Self-attention with relative position representations", https://www.aclweb.org/anthology/N18-2074.pdf) in place of the usual Transformers based on sinusoidal positional encodings, and hope to try this in future work; this submission will provide a strong baseline to evaluate such future work. We thank the reviewer for raising this point. We now include this possibility as a future extension in Section 5, which we rename from “Conclusions” to “Conclusions and Future Work”.
>
> With regard to types, we do not assume that the source code is written in a statically typed language and hence, do not use types as features. We train CuBERT for Python code, which is dynamically typed. We leave exploring type information for statically-typed languages for future work.
>
> >> Burden of program analysis
>
> The reviewer’s point is well taken. We have reworded our relevant text in the paper. To recap, our goal is to understand and evaluate a BERT-like pre-training approach (i.e., purely on lexical information) for program-understanding tasks, without exposing to the pre-training model additional information gleaned through program analysis.
>
> >> Simplicity of the Function-Docstring Mismatch task
>
> We agree with the reviewer’s observation. Nevertheless, the inherent ability of Transformer to relate every pair of tokens through self-attention plays a crucial role in CuBERT getting +7.5% improvement over the baseline model even on this relatively simple task.
>
> >> Utility of Next Sentence Prediction task
>
> A recent work has argued that using only the Masked Language Model objective (coupled with more training on larger datasets) can improve the performance of BERT (see "RoBERTa: An optimized method for pretraining self-supervised NLP systems", https://arxiv.org/abs/1907.11692). It will take more experimentation to check how inclusion/exclusion of next-sentence-prediction affects CuBERT.
>
> >> Explanation of similarity metric
>
> Two files are considered similar to each other if the Jaccard similarity between the sets of tokens (identifiers and string literals) is above 0.8 and in addition, it is above 0.7 for multi-sets of tokens. This is based on the criteria used in Allamanis (2018). We have added this explanation in the revised version.
>
> >> Fraction of positive/negative examples in the finetuning tasks
>
> We have provided these details in Appendix A now. To summarize, all classification tasks except Exception Type classification, and the new pointer task (Section 4.7), have a 50-50 split of buggy/bug-free examples. The per-class counts for the Exception Type classification task are shown in the (new) Table 6.
>
> >> Motivation for making Variable Misuse and Wrong Operator/Operand as classification tasks instead of correction tasks
>
> We have now included experimentation for the joint task of classification, localization and repair of variable misuse errors from Vasic et al. (2019). Please see Section 4.7 for the results. The original Wrong Operator and Swapped Operand tasks from (Pradel & Sen 2018) are binary classification tasks similar to ours. They were not error correction tasks.

---

> > ### Author Response · Authors · 2019-11-15
> > **Follow-up**
> >
> > Dear reviewer #3,
> >
> > We hope that our response addresses your comments. Please let us know if you need any additional clarifications.
> >
> > Thank you for your review.

---

> > > ### Comment · AnonReviewer3 · 2019-11-15
> > > **Thank you**
> > >
> > > Thank you for the answers and clarifications. I read the new draft, and it addresses my concerns much better — I particularly appreciated addition of a more complex version of the VarMisuse task.
> > >
> > > My recommendation will stay at "Weak Accept" as (a) the paper's modeling contributions are relatively weak, as opposed to pretraining/empirical contributions, and (b) as an application of BERT to the program analysis domain, it does not yet make sufficient use of the information available in the domain. That said, I agree with the authors that it could be left out of the scope of this paper, and will argue for acceptance nonetheless.

---

### Author Response · Authors · 2019-11-13
**Paper updated**

We thank the reviewers for their detailed comments and suggestions. We are responding to the individual comments below.

We have updated our writeup as follows:
1) We have added a more complex finetuning task for joint classification, localization and repair of variable-misuse errors (based on Vasic et al. 2019), which we compare to the multi-headed pointer model from Vasic et al. (2019), trained and evaluated on our datasets (Section 4.7). The results show that CuBERT performs well on this task also.
2) We have added an appendix (Appendix A) giving details about data generation for the finetuning tasks and include a discussion on the careful use of pseudorandomness towards dataset reproducibility.
3) We have added clarifications and discussed possible extensions in the paper based on the reviewers’ suggestions.

We would be happy to answer any additional queries.

---

### Decision · Program_Chairs · 2019-12-19

**Decision:**

Reject

**Comment:**

The paper presents  CuBERT (Code Understanding BERT), which is BERT-inspired pretraining/finetuning setup, for source code contextual embedding. The embedding results are tested on classification tasks to demonstrate the effectiveness of CuBERT.

This is an interesting application paper that extends existing models to source code analysis. The authors did a good job at motivating the applications, describing the proposed models and discussing the experiments. The authors also agree to share all the datasets and source code so that the experiment results can be replicated and compared with by other researchers.

One major concern is the lack of strong baselines. All reviewers are concerned about this issue. The paper could lead to a good publication in the future if the issues can be addressed.

---

> ### Author Response · Authors · 2019-12-20
> **Regarding strong baselines**
>
> The focus of our paper has been to study the effectiveness of the BERT-style pre-training for source code and to bring the latest advances in the NLP field around pre-training to the ML-for-programming field. We therefore compared fine-tuning of CuBERT against end-to-end training (of both BiLSTM and Transformer models) and training with learned Word2Vec embeddings. These were the baselines we compared against.
>
> As argued in the paper and subsequently in the rebuttal phase, even though our fine-tuning tasks are motivated by similar ones in the literature, those papers have used different languages or different formulations of the tasks, making it infeasible to directly compare against them. During the discussion, this was also acknowledged by the reviewer(s) as orthogonal to our contributions. Nevertheless, as suggested by two reviewers during rebuttal, we added an additional task, that of variable misuse localization and repair, and compared against the SOTA model from prior work.
>
> While we respect the decision of the reviewers/AC/PC on the paper, we find the rationale for the rejection (lack of strong baselines) to be incongruous with the reviews and the entire discussion.